# Optimal planning and partitioning of multiple distribution Micro-Grids based on reliability evaluation

**Hamid Amini Khanavandi, Majid Gandomkar** **\*, Javad Nikoukar**

Department of Electrical Engineering, College of Engineering Technology, Saveh Branch, Islamic Azad University, Saveh, Iran

\* majid.gandomkar@iau.ac.ir

## Abstract

Power system researcher have turned to Micro-Grids (MG) for higher reliability, greater flexibility, lower operating costs and losses, and lower $CO_2$ emissions at the distribution system. This paper presents a single-level stochastic optimization framework for planning and partitioning of a distribution system including Multiple Micro-Grids (MMGs). The main objective is to minimize the total cost of the system including investment, operation, total losses and reliability costs of the distribution network. The proposed model takes into account the viewpoints of MG owners and distribution system operators, simultaneously. The voltage stability index is introduced to identify the optimal site of MG investment. To deal with uncertainties caused by renewable generations, the Firefly Algorithm (FA) and probability-tree method is used to create various operation scenarios of Photo-Voltaic (PVs) and Wind Turbines (WTs). This model is solved through the genetic algorithm in MATLAB, and to evaluate its effectiveness, numerical studies have been carried out on the experimental IEEE distribution network with four specified locations for investing MGs and seven Tie Switches (TS) for network partitioning. Simulation results reveal that optimal locations for MG investment are determined in such a way all MGs connect to the buses near the beginning of the feeder and as a result, load point reliability is improved, total active power losses are reduced, and the energy program becomes more optimized.

## 1. Introduction

Micro-Grid (MGs) technology is a new emerging concept in the power system society. According to recent developments, the application of various types of MGs has become more popular in modern power systems. The appearance of MGs could especially bring many benefits to the distribution systems including higher reliability and flexibility, less operation and loss costs, and lower emission [1, 2].

Reference [3] investigates a method for clustering the traditional passive distribution system in a cluster of MMGs network. Among the benefits of traditional distribution system clustering in MMG for customers and distribution system operators include local control strategy to

**Competing interests:** The authors have declared that no competing interests exist.

**Abbreviations:** $P_{ti}^{MGi(CHPi)}$, Active power generation of $CHPi$ in $MGi$ at the operation time period $ti$; $P_{ti}^{MGi(PVi)}$, Active power generation of $PVi$ in $MGi$ at the operation time period $ti$; $P_{ti}^{MGi(WTi)}$, Active power generation of $WT_i$ in $MGi$ at the operation time period $ti$; $P_{Bi,ti}^{MGi(Shed)}$, Amount of load shedding from Bi in $MGi$ at operation time period $ti$; $k_{ec}^{MGi}$, Asset lifetime of {DERs, ESSs and DRSs} in $MGi$; $k_{TS}^{DN}$, Asset lifetime of TS in the distribution network; $H_{ave,ti}^{Sun}$, Average severity of sun radiation scenario at the operation time period $ti$; $Voll_e^{MGi}$, Average value of lost load for consumers in $MGi$; $b_{Ch,ti}^{MGi(EESi)}$, Binary variables that indicate to charge of $ESSi$ in $MGi$ at the operation time period $ti$; $b_{Disch,ti}^{MGi(EESi)}$, Binary variables that indicate to discharge of $ESSi$ in $MGi$ at the operation time period $ti$; $P_{e,ti}^{MGi(Buy)}$, Bought energy for $MGi$ from distribution network at the operation time period $ti$; $r_{line}$, Branch resistance; $I_{ti}^{MGi(linei)}$, Current flow of $Linei$ in $MGi$ at the operation time period $ti$; $I_{ti}^{linei}$, Current flows through the branch $linei$ at operation time period $ti$; $\pi_{e,ti}^{MGi(Buy)}$, Energy buying price into the power market for $MGi$ at the operation time period $ti$; $\pi_{e,ti}^{MGi(Sell)}$, Energy selling price into the power market for $MGi$ at the operation time period $ti$; $P_{ti}^{MGi(EESi)}$, ESS's charging/discharging power of $ESSi$ in $MGi$ at the operation time period $ti$; $SoC_{ti}^{MGi(EESi)}$, ESS's stored energy of $ESSi$ in $MGi$ at the operation time period $ti$; $C_{O,ti}^{TotLoss,Ex}$, Expected cost of active power loss in distribution system at the operation time period $ti$; $C_{O,ti}^{MGi(Buy\&sell),Ex}$, Expected cost of trading with upstream distribution network for $MGi$ at the operation time period $ti$; $C_{O,ti}^{MGi(EENS)}$, Expected energy not served of consumers for $MGi$ at the operation time period $ti$; $C_{O,ti}^{MGi(DERs),Ex}$, Expected operating costs of DERs for $MGi$ at the operation time period $ti$; $C_O^{DN,Ex}$, Expected operation costs of distribution system; $IP_{Cap\_Max}^{MGi(CHPi)}$, Installation capacities of CHP; $IP_{Cap\_Max}^{MGi(ESSi)}$, Installation capacities of ESS; $IP_{Cap\_Max}^{MGi(PVi)}$, Installation capacities of PV; $IP_{Cap\_Max}^{MGi(WTi)}$, Installation capacities of WT; $IC_{SW}^{Tie}$ Installation cost of a TS in the distribution network; $P_{ec}^{max}$, Installed capacity of {DERs, ESSs and DRSs} in $MGi$; $r_{ec}^{MGi}$, Interest rate of {DERs, ESSs and DRSs} in $MGi$; $r_{TS}^{DN}$, Interest rate of TS in the distribution network; $\eta_{pvi}^{Inv}$, Inverter efficiency of PV; $N_{SW}^{MAX}$, Maximum numbers of TSs; $OC_{ti}^{MGi(CHPi)}$, Operating cost of $CHPi$ in $MGi$ at the operation time period $ti$; $OC_{ti}^{MGi(CHPi)}$, Operating costs of CHPi in MGi at the operation time period $ti$; $N_{TSS}^{Opt}$ Optimal number of tie TS in the distribution network; $N_{TSS}^{Branch}$, Optimal numbers of TSs; $P_{ti}^{MGi(WTi)}$, Power generation of $WTi$ in $MGi$ at the operation time period $ti$; $Q_{ti}^{MGi(CHPi)}$, Reactive power generation of $CHPi$ in

minimize interaction between different MGs, local reactive power compensation to prevent fault propagation, minimize power losses and ultimately improve network reliability. Reference [4] also provides a suitable model for strengthening the security of MMG-based active distribution networks by optimally using consumption management plans and energy storage resources. In the proposed model, a hierarchical two-stage approach has been established in the form of an incident and its impact on the distribution network and preventive and corrective measures to increase system readiness and reduce damages caused by severe accidents.

It is expected that distributed energy resources such as solar Photo-Voltaic (PVs) and Wind Turbines (WTs) play an important role in the electricity supply markets and low carbon economy shortly. Nevertheless, the high utilization of these renewable resources in MGs has caused new challenges from both operation and planning perspectives. Usually, MGs operate in isolated or grid-connected modes. The selection of the optimal operation mode for MGs in the distribution system using the posture of Tie Switches (TSs) is another challenge [5].

Due to PVs and WTs power generation uncertainty, the Multiple MGs (MMGs) planning process has converted to a complex nonlinear optimization problem that needs a powerful method to be solved optimally [6].

A parallel computing framework based on spectral partitioning aimed at increasing the computational efficiency of large-scale microscopic traffic simulation is introduced in [7]. This framework uses the SP method to segment road networks, considering vehicle information and road information as building blocks to determine node weights.

MMGs planning process has two phases. The primary phase is DERs optimal placement in MGs and the secondary phase is TSs optimal alignment between MGs and also MGs with the distribution system. The mentioned problem has been modelled using deterministic and probabilistic methods. Probabilistic methods can be divided into stochastic and robust forms when system uncertainties are considered. Up to now, many studies have been conducted on the planning and partitioning MMGs about the various objectives and solution algorithms [8, 9].

Probabilistic indices for analyzing the impact of distributed energy resource penetration on system reliability have been surveyed in Reference [10]. A scenario-based investment planning of isolated MMGs is presented in [11] considering electricity, heating and cooling demands. The topology of flexible interconnected distribution network is analyzed according to [12] and the concept of partition and block is proposed. The flexible interconnected distribution network of this reference is divided into three areas: DC, AC and SOP. Based on the partition, Dijkstra's algorithm is used to search for the shortest circuit to form the block, and the block information of each area is specified.

Reference [13] developed an expansion planning model using the bender decomposition method. It takes into account the trade-off between MGs installation cost in distribution networks and generation-transmission expansion investment cost. Reference [14] presented a new bi-level model for MGs planning considering load consumption and PV and WTs power generation uncertainties. Monte Carlo simulations are used to create operating scenarios. Decision-making is performed by the distribution system operator and the MGs owners' mutual interaction.

A multi-objective mean-semi-entropy model was proposed in reference [15] for an off-grid MG including PVs, WTs, ESSs and DGs through the economic benefits and reliability cost evaluation for renewable energy resources. An interval optimization tool for planning MMGs is presented in [16] that selects the optimal site and size of equipment, and corresponding economic dispatches about operation uncertainties. A comprehensive model for off-grid MGs design is proposed in Reference [17] that has various types of DGs shunt capacitor banks and nonlinear loads. The harmonic power flow tool is tailored for planning goal that takes into account the mentioned specific features of off-grid MGs.

$MGi$ at the operation time period ti; $Q_{ti}^{MGi(DRSi)}$, Reactive power of $DRSi$ in $MGi$ at the operation time period ti; $P_{e,ti}^{MGi(Sell)}$, Sell energy to distribution network from $MGi$ at the operation time period ti; $V_{ti}^{MGi(Busi)}$, Voltage magnitude of $Busi$ in $MGi$ at the operation time period ti; $A_{PV}$, area of PV panel; $BESSs$, Battery Energy Storage Systems; $CHP$, Combined Heat and Power; $DERs$, Distributed Energy Resources; $DGs$, Distributed Generators; $DRSs$, Distributed Reactive Sources; $DSO$, Distribution System Operator; $IC_{ec}$, Unit cost of {DERs, ESSs and DRSs} for installation a new capacity; $k_e$, Cost of energy loss in term of dollar per KWh; $MMGs$, Multiple Micro Grids; $N_{PVsc}$, Number of PV panel operation scenarios; $N_{PVsc}$, Number of PVs operation scenarios; $N_{WTsc}$, Number of WTs operation scenarios; $P_r$, Rated power of WT; $PV$, Photo-Voltaic; $Sci$, Operation scenarios related to PVs; $Scj$, Operation scenarios related to WTs; $S^{MGi(CHPi)}$, Apparent power of $CHPi$ in $MGi$; $TC$, Total cost of MMGs in distribution system in the planning horizon; $TSs$, Tie Switches; $v_{ci}, v_{co} v_r$, Cut-in, cut-out and rated of wind speed, respectively; $VSI$, Voltage Stability Index; $v_{w,ti}$, wind speed at the operation time period ti; $WT$, Wind Turbine; $\alpha_{CHPi}, \beta_{CHPi}, \gamma_{CHPi}$, Cost function coefficient of $CHPi$; $\rho(Sci, Scj)$, Occurrence probability of $(Sci, Scj)$ operation scenarios related to PVs and WTs, simultaneously.

Reference [18] has presented a multi-stage model to schedule MGs when their interconnection with the distribution system can be opened. To reduce the adverse effects of uncertain renewable energy output, a distributed robust bi-level model predictive control-based energy management strategy is proposed in References [19] for off-grid MMGs. A decentralized bi-level energy management model was discussed in reference [20] for the coordinated operation of networked MGs in a distribution system. A bi–level model for a coupled MGs energy and reserve capacity planning process was proposed in Reference [21], upper level aims for the minimization of MG planning and operational costs, while the lower level guarantees a reliable power supply. In reference [22], a bi-level planning model based on the game theory method has been developed which considers operation risk under long-term competitive power market situations and distribution system operators. For validation, it has been performed on a 33-bus IEEE standard distribution network [23]. Also, in references [24, 25], a method has been used to improve the performance of renewable resources and reduce their production power fluctuations.

To control the risk of renewable resources with high volatility, a risk-based model for optimal planning and segmentation of MMGs considering the uncertainty of renewable resources is presented in [26]. In addition, the term techno-economic risk has been introduced to analyze the total risk in MMG. Also, the energy sources of distributed generation and storage in MGs and the state of connection switches to connect MGs to the network are determined simultaneously.

A bi-level optimized operation strategy for the MG cluster was described in References [27] which aimed to improve their economic benefits and reduce their operating risk. As can be seen in review papers [18–23, 27], all of them cause suboptimal solutions for planning and operation planning of MGs in the distribution system due to application of the multi-step algorithm. To achieve the best model for MMGs planning and partitioning problems, References [22, 23, 27] are considered as the base papers for this research. Our paper proposes a reliability-based model in which the optimal sites and sizes of DERs, DRSs and ESSs in MMGs and also TSs placement in the distribution network have been determined, simultaneously. In references [28–30], methods for estimating the life of electrical network equipment have been examined.

At present, there is few literature to establish a reliability-based model about the expected costs of operation, risk and total active power loss besides planning the cost of MGs in the distribution network, but this paper considers these goals in a single-level model. The optimal location of MGs for investment in DERs, DRS and BESS is determined based on the voltage stability index (VSI) criterion. Because, the proper location for MGs in the distribution network can change the situation of system reliability, voltage profile and stability. Variable output power of PVs and WTs renewable resources are defined as MGs operation uncertainties and the probability-tree method is used to create all operation uncertainties. Since, the MMGs planning and partitioning optimization problem is inherently non-linear, for achieving a solution near to global optima, Firefly Algorithm (FA) is employed and also our solution method for the proposed model is superior in this research as compared to others. So, by reviewing the references and stating the goals of each one, the summary of the work done and its comparison with the innovations of this paper is shown in Table 1.

Remain of the paper is arranged as follows: in section 2, a mathematical model for planning and partitioning of MGs has been formulated based on reliability and economic considerations. The solution flowchart and mentioned problem coding procedure into FA have been presented in section 3. Numerical studies and simulations are implemented in section 4. In section 5, achieved results through FA are compared to other well-known methods such as

  

**Table 1. Compare the proposed method and previous methods.**

| No. | Aspect | [3] | [4] | [18] | [19] | [22] | [23] | [24] | [25] | [26] | [27] | [28] | [29] | [30] | This paper |
|---|---|---|---|---|---|---|---|---|---|---|---|---|---|---|---|
| 1 | Partitioning | ✓ | ✓ | | | | | | | ✓ | | | | | ✓ |
| 2 | Uncertainty | | ✓ | ✓ | ✓ | ✓ | ✓ | ✓ | ✓ | | | | | | ✓ |
| 3 | Equipment Aging | | ✓ | | | | | | | | | ✓ | ✓ | ✓ | |
| 4 | DM Algorithm | | | | | | ✓ | | | | | ✓ | ✓ | ✓ | |
| 5 | MG Clustering | ✓ | ✓ | | | | | | | | ✓ | | | | |
| 6 | Opt. Algorithm | | ✓ | ✓ | | | ✓ | | | | ✓ | | | | ✓ |
| 7 | MMG | ✓ | ✓ | | ✓ | | ✓ | | | ✓ | | | | | ✓ |
| 8 | Prediction | | | ✓ | ✓ | | | ✓ | ✓ | | | | | | ✓ |
| 9 | Reliability | ✓ | ✓ | | | | | | | | | | | | ✓ |
| 10 | FA | | | | | | | | | | | | | | ✓ |
| 11 | expected costs of operation | | | | | | | | | | | | | | ✓ |
| 12 | VSI | | | | | | | | | | | | | | ✓ |
| 13 | probability-tree method | | | | | | | | | | | | | | ✓ |

DM: Decision-Making MG: Micro-Grid Opt.: Optimization

MMG: Multi Micro-Grid FA Firefly Algorithm VSI: Voltage Stability Index

particle swarm optimization and genetic algorithm for all case studies and conclusion remarks have been expressed in section 6.

# 2. Mathematical model for planning and partitioning of MGS

In this section, mathematical model for planning and pertaining based on objective function and technical constraints will be explained. Fig 1 shows the structure of the system investigated in this paper. According to this figure, the components of this system are shown to be connected to the MGs.

Also, Fig 2 shows the overall procedure of solving the proposed problem of this article. In the following, each of the steps will be explained in detail.

## 2.1 Objective function

The main objective is to minimize the total cost of the system including investment, operation, total losses and reliability costs of the distribution network. Mathematical modelling of MMGs Planning and partitioning includes the objective function and technical-economic constraints. Here, the objective function is defined as (1).

$$Min\left(of_{Planning}^{MMGs} = TC\right) \rightarrow TC = \left\{\sum_{MGi=1}^{Nmg} C_{Inv}^{MGi} + \sum_{SWi=1}^{Nsw} C_{Inv}^{SWi}\right\} + C_O^{DS,Ex} \tag{1}$$

In the objective function, there are two terms. The first one is the sum of investment costs for all MGs and TSs in the distribution system, and the second one is the expected operation costs of the distribution system considering various operating scenarios. The investment cost

  

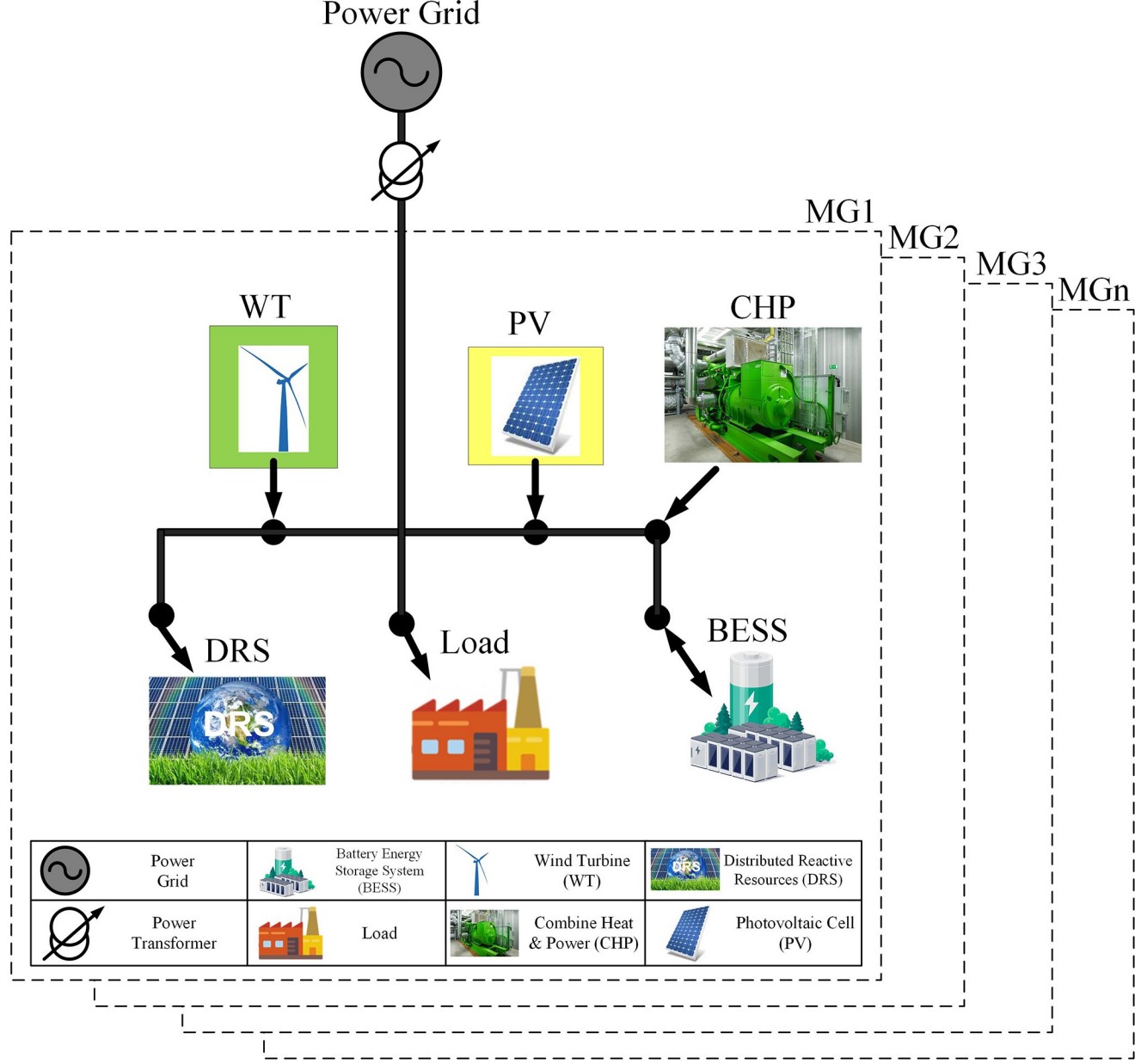

**Fig 1. System structure.**

of MG is calculated using (2), (3) and (4).

$$C_{Inv}^{MGi} = \sum_{ec=1}^{EC} \frac{r \times \left(1 + r_{ec}^{MGi}\right)^{k_{ec}^{MGi}}}{\left(1 + r_{ec}^{MGi}\right)^{k_{ec}^{MGi}} - 1} \times C_{ec}^{MGi} \tag{2}$$

$$C_{ec}^{MGi} = IC_{ec} P_{ec}^{\max} \rightarrow \forall ec \in \Omega_{MGs} \tag{3}$$

$$\Omega_{MGs} = \{PVs, WTs, CHPs, ESSs, DRSs\} \tag{4}$$

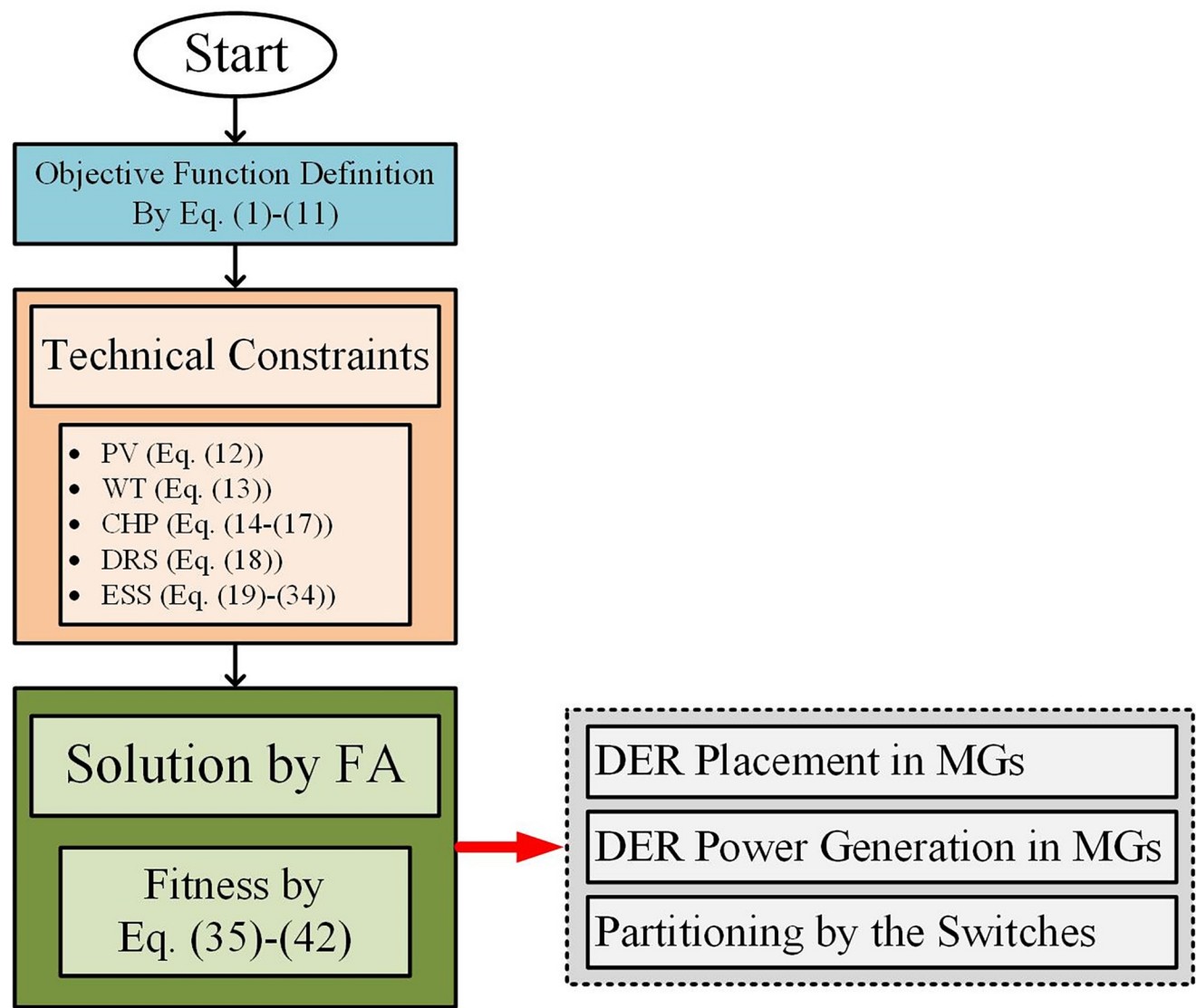

**Fig 2. Overall procedure of solving the proposed problem.**

The interest rate and asset lifetime of DERs, ESSs and DRSs are applied to express the financial value of installation expenditure based on the first installation year. Similarly, the investment cost on TS for partitioning goals can be calculated using (5) and (6).

$$C_{Inv}^{SWi} = \frac{r \times \left(1 + r_{TS}^{DN}\right)^{k_{TS}^{DN}}}{\left(1 + r_{TS}^{DN}\right)^{k_{TS}^{DN}} - 1} \times C_{TS}^{DN} \tag{5}$$

$$C_{TS}^{DN} = IC_{SW}^{Tie} \times N_{TSs}^{Opt} \tag{6}$$

Here, one year is divided into four seasons and also each period for operation planning study is equal to three months. The expected cost of the distribution network in each operation

planning period can be computed according to relation (7).

$$C_{O,ti}^{DN,Ex} = C_{O,ti}^{TotLoss,Ex}$$
$$+ \sum_{MGi=1}^{Nmg} (C_{O,ti}^{MGi(EENS)} + C_{O,ti}^{MGi(DERs),Ex} + C_{O,ti}^{MGi(Buy \& sell),Ex}) \tag{7}$$

Variable output power of PVs and WTs renewable resources in MGs are given as operation uncertainties. Therefore, the expected operating cost of the distribution network consists of four expected values. The first term is the expected cost of active power loss in the distribution system, second and third terms are expected energy not served by consumers and expected operating costs of DERs for all MGs, respectively. The fourth term is the expected cost of trading with the upstream distribution network for all MGs. The expected cost of active power loss in the distribution system at each operation period is calculated using (8).

After AC power flow study under various operation scenarios of PVs and WTs renewable resources in the distribution network, the Expected cost of active power loss can be calculated using given currents flow through all branches during each load level. Expected energy not served can be calculated as a (9). In our model, the expected operating costs of ESSs, DRSs and SWs are approximately assumed equal to zero, and therefore expected operating costs of DERs in all MGs are limited to CHPs activation, and it can be calculated according to (10). The expected cost of trading between an upstream distribution networks can be modelled according to (11).

$$C_{O,ti}^{TotLoss,Ex} = \sum_{SCi=1}^{N_{PVsc}} \sum_{SCj=1}^{N_{WTsc}} (\rho(Sci, Scj) \times k_e \times t_i \times \sum_{line_i=1}^{Nl} (r^{line_i} \times |I_{ti}^{line_i}(Sci, Scj)|^2)) \tag{8}$$

$$C_{O,ti}^{MGi(EENS)} = \sum_{SCi=1}^{N_{PVsc}} \sum_{SCj=1}^{N_{WTsc}} (\rho(SCi, SCj) \times t_i \times \sum_{B_i=1}^{NB} (Voll_e^{MGi} \times P_{Bi,ti}^{MGi(Shed)}(SCi, SCj))) \tag{9}$$

$$C_{O,ti}^{MGi(DERs),Ex} = \sum_{SCi=1}^{N_{PVSC}} \sum_{SCj=1}^{N_{WTSC}} (\rho(SCi, SCj) \times t_i \times \sum_{CHP_i=1}^{Nchp} OC_{ti}^{MGi(CHPi)}(SCi, SCj)) \tag{10}$$

$$C_{O,ti}^{MGi(Buy\&sell),Ex} = \sum_{SCi=1}^{N_{PVSC}} \sum_{SCj=1}^{N_{WTSC}} (\rho(SCi, SCj) \times t_i \times \left\{ \begin{array}{l} (\pi_{e,ti}^{MGi(Buy)} \times P_{e,ti}^{MGi(Buy)}(SCi, SCj) \\ -(\pi_{e,ti}^{MGi(Sell)} \times P_{e,ti}^{MGi(Sell)}(SCi, SCj)) \end{array} \right\}) \tag{11}$$

## 2.2 Technical constraints

There are different types of DERs, ESSs and DRSs in each MGs that should be modelled before technical constraints have been formulated. In this section, Technical constraints of MMGs planning and partitioning are modelled including permissible active and reactive power generation of DERs, bus voltage limit, thermal limit of branches, active and reactive power balance between generation and demand under normal and contingencies conditions, maximum allowable number of TSs and maximum allowable current flow through TSs.

**2.2.1 PV panel and WT farm.** Output powers of the PV panel and WT in each MG are modelled according to (12) and (13), respectively.

$$P_{ti}^{MGi(PVi)}(sc_{pvi}) = \eta_{pvi}^{Inv} \times A_{PV} \times H_{ave,ti}^{Sun}(sc_{pvi}) \tag{12}$$

$$P_{ti}^{MGi(WTi)}(sc_{wtj}) = \begin{cases} 0 \rightarrow v_{w,ti}(sc_{wtj}) < v_{ci} \ and \ v_{w,ti}(sc_{wtj}) > v_{co} \\ P_r \dfrac{v_{w,ti}(sc_{wtj}) - v_{ci}}{v_r - v_{ci}} \rightarrow v_{ci} < v_{w,ti}(sc_{wtj}) < v_r \\ P_r \rightarrow v_r < v_{w,ti}(sc_{wtj}) < v_{co} \end{cases} \tag{13}$$

**2.2.2 CHP units.** The fuel cost of a CHP unit in each MG under specified wind speed and sun radiation scenarios is modelled using (14).

$$OC_{ti}^{MGi(CHPi)}(sc_{pvi}, sc_{wtj}) = \alpha_{CHPi} + \beta_{CHPi} \times P_{ti}^{MGi(CHPi)}(sc_{pvi}, sc_{wtj})$$

$$+ \gamma_{CHPi} \times \left( P_{ti}^{MGi(CHPi)}\left(sc_{pvi}, sc_{wtj}\right) \right)^2 \tag{14}$$

The CHP active and reactive power generation limits and their interdependency in MG under specified wind speed and sun radiation scenarios are calculated using (15), (16) and (17).

$$P_{\min}^{MGi(CHPi)} \leq P_{ti}^{MGi(CHPi)}(sc_{pvi}, sc_{wtj}) \leq P_{\max}^{MGi(CHPi)} \tag{15}$$

$$Q_{\min}^{MGi(CHPi)} \leq Q_{ti}^{MGi(CHPi)}(sc_{pvi}, sc_{wtj}) \leq Q_{\max}^{MGi(CHPi)} \tag{16}$$

$$\sqrt{P_{ti}^{MGi(CHPi)^2}\left(sc_{pvi}, sc_{wtj}\right) + Q_{ti}^{MGi(CHPi)^2}\left(sc_{pvi}, sc_{wtj}\right)} \leq S^{MGi(CHPi)} \tag{17}$$

**2.2.3 DRS units.** Reactive power generation of DRS Units in each MG under specified wind speed and sun radiation scenarios is modelled according to relation (18).

$$Q_{\min}^{MGi(DRSi)} \leq Q_{ti}^{MGi(DRSi)}(sc_{pvi}, sc_{wtj}) \leq Q_{\max}^{MGi(DRSi)} \tag{18}$$

**2.2.4 ESSs units.** The stored energy of the ESS unit depends on power generation surplus or shortage scenarios in each MG which is modelled using (19). Injected/absorbed power with ESS to/from the distribution system can be calculated according to (20). ESS's charging and discharging power is modelled using (21). Charge and discharge of ESS in an MG do not

simultaneously occur that can be modelled by (22).

$$SoC_{Min}^{MGi(EESi)} \leq SoC_{ti}^{MGi(EESi)}(sc_{pvi}, sc_{wtj}) \leq SoC_{Max}^{MGi(EESi)} \tag{19}$$

$$P_{ti}^{MGi(EESi)}(sc_{pvi}, sc_{wtj}) = \eta_{EES}$$
$$\times \left[ SoC_{ti}^{MGi(ESSi)}(sc_{pvi}, sc_{wtj}) - SoC_{ti}^{MGi(ESSi)}(sc_{pvi}, sc_{wtj}) \right] \tag{20}$$

$$b_{Disch,ti}^{MGi(EESi)}(sc_{pvi}, sc_{wtj})$$
$$\times P_{disc,EES}^{Max} \leq P_{ti}^{MGi(EESi)}(sc_{pvi}, sc_{wtj}) \leq b_{Ch,ti}^{MGi(EESi)}(sc_{pvi}, sc_{wtj}) \times P_{Ch,EES}^{Max} \tag{21}$$

$$b_{Disch,ti}^{MGi(EESi)}(sc_{pvi}, sc_{wtj})$$
$$+ b_{Ch,ti}^{MGi(EESi)}(sc_{pvi}, sc_{wtj}) \leq 1 \rightarrow \left\{ b_{Disch,ti}^{MGi(EESi)}, b_{Ch,ti}^{MGi(EESi)} \right\} \in [0, 1] \tag{22}$$

Power generation constraints for DERs including PVs, WTs, CHPs, ESSs and DRSs in each MG are modelled using (23), (24), (25), and (26), respectively.

$$P_{ti}^{MGi(PVi)}(sc_{pvi}) \leq IP_{Cap\_Max}^{MGi(PVi)} \tag{23}$$

$$P_{ti}^{MGi(WTi)}(sc_{wtj}) \leq IP_{Cap\_Max}^{MGi(WTi)} \tag{24}$$

$$P_{ti}^{MGi(CHPi)}(sc_{pvi}, sc_{wtj}) \leq IP_{Cap\_Max}^{MGi(CHPi)} \tag{25}$$

$$P_{ti}^{MGi(ESSi)}(sc_{pvi}, sc_{wtj}) \leq IP_{Cap\_Max}^{MGi(ESSi)} \tag{26}$$

Reactive power generation of DRS and CHP in MG has been modelled using (27) and (28), respectively.

$$Q_{ti}^{MGi(CHPi)}(sc_{pvi}, sc_{wtj}) \leq IQ_{Cap\_Max}^{MGi(CHPi)} \tag{27}$$

$$Q_{ti}^{MGi(DRSi)}(sc_{pvi}, sc_{wtj}) \leq IQ_{Cap\_Max}^{MGi(DRSi)} \tag{28}$$

Under each operation scenario of the distribution system, voltage magnitude for all buses and current for all branches in MGs should be in permissible values according to (29) and (30).

$$V_{Min}^{Bus} \leq V_{ti}^{MGi(Bus_i)}(SCi, SCj) \leq V_{Max}^{Bus} \tag{29}$$

$$I_{Branch-Min}^{MGi} \leq I_{ti}^{MGi(line_i)}(SCi, SCj) \leq I_{Branch-Max}^{MGi} \tag{30}$$

Under each operating condition, active and reactive power generation and consumption in

all MGs should be equal according to (31) and (32).

$$P_{e,ti}^{MGi(Buy)}(SCi, SCj) + \sum_{\substack{SCi, SCj \\ SCi = \{sc_{pv1}, sc_{pv2} sc_{pvn}\} \\ SCj = \{sc_{wt1}, sc_{wt2} sc_{wtm}\}}} \left( \sum_{DERi=1}^{N_{DER}} P_{ti}^{MGi(DERi)}(SCi, SCj) \right.$$

$$\left. + \sum_{ESSi=1}^{N_{ESS}} P_{Disch,ti}^{MGi(ESSi)}(SCi, SCj) \right) = \sum_{\substack{SCi, SCj \\ SCi = \{sc_{pv1}, sc_{pv2} sc_{pvn}\} \\ SCj = \{sc_{wt1}, sc_{wt2} sc_{wtm}\}}} \left( \sum_{ESSi=1}^{N_{ESS}} P_{Ch,ti}^{MGi(ESSi)}(SCi, SCj) + \sum_{Bi=1}^{N_{Bi}} P_{Bi,ti}^{MGi(di)} \right)$$

(31)

$$\sum_{\substack{SCi, SCj \\ SCi = \{sc_{pv1}, sc_{pv2} sc_{pvn}\} \\ SCj = \{sc_{wt1}, sc_{wt2} sc_{wtm}\}}} \left( \sum_{\substack{DERi=1 \\ DERi = \{CHPi\}}}^{N_{DER}} Q_{ti}^{MGi(DERi)}(SCi, SCj) \right.$$

$$\left. + \sum_{DRSi=1}^{N_{DRS}} Q_{ti}^{MGi(DRSi)}(SCi, SCj) \right) = \sum_{Bi=1}^{N_{Bi}} Q_{Bi,ti}^{MGi(di)}$$

(32)

Current flowing through TSs and the maximum number of TSs for the installation in distribution system have been modelled according to (33) to (34) based on the available budget for the distribution network operator, respectively.

$$I_{TSs,ti}^{Branch}(SCi, SCj) \le I_{Branch}^{MAX}$$

(33)

$$\sum_{sw_i=1}^{N_{SW}} N_{TSs}^{Branch} \le N_{SW}^{MAX}$$

(34)

## 3. Solution flowchart and problem coding procedure in FA

Here, the solution flowchart and coding method of MGs planning and partitioning problem in FA have been discussed. The initial population of fireflies are generated, randomly. According to Table 2, location of each firefly is equal to the problem solution. DERs placement in all MGs under the planning time horizon, DERs economic dispatch under the operation planning time horizon, and optimal number and location of TSs for partitioning of MGs are independent variables in the proposed model. Inputs to the model are capacity limits for DERs, weather information in the region for power prediction of PVs and WTs renewable resources, load demand, network impedance information and available budget for TSs installation by goal of MMGs partitioning.

Usually, Voltage collapse occurs in some buses of the distribution network when customers' active load demand increase to maximum. Here, optimal locations of MGs for investment in the distribution system have been determined from candidate locations using VSI. Buses with lower values of VSI are more susceptible to voltage collapse, thus MGs investment in them is vital. VSI calculation is illustrated by the use of Fig 3.

**Table 2. A typical location of FA is equal to problem solution.**

| Distribution System (DER placement in MGs) | | | Distribution System (DER power generation in MGs) | | | Distribution System (Partitioning by tie switches) |
|---|---|---|---|---|---|---|
| MG1....MGn | | | MG1....MGn | | | MG1....MGn |
| CHP1..CHPn | DRS1..DRSn | ESS1..ESSn | CHP1..CHPn | DRS1..DRS | ESS1..ESSn | SW1..sWk |
| $IP_{CHP}(Bus1..Busm)$ | $P_{DRS}(Bus1..Busm)$ | $IP_{ESS}(Bus1..Busm)$ | $PG_{CHP}(Bus1..Busm)$ | $PG_{DRS}(Bus1..Busm)$ | $PG_{ESS}(Bus1..Busm)$ | $Location_{SW}(Branch1..Branchk)$ |

After AC power flow, the mathematical formulation of VSI can be expressed as below (35).

$$
\begin{aligned}
VSI_j(n) = |V_i(n)|^4 &- 4\left\{P_j^{total}(n).X_{ij}(n) - Q_j^{total}(n).R_{ij}(n)\right\}^2 \\
&- 4\left\{P_j^{total}(n).R_{ij}(n) - Q_j^{total}(n).X_{ij}(n)\right\}|V_i(n)|^2
\end{aligned}
\tag{35}
$$

Indexes $P_j^{total}(n)$ and $Q_j^{total}(n)$ are total active and reactive powers fed to load at receiving end of branch n can be simply computed for bus j from 2 to M. Index $VSI_j(n)$ is voltage stability index for ending bus of branch n. Indexes $V_i(n)$ and $V_j(n)$ are voltage magnitude of sending and ending buses of branch n, respectively. Indexes $R_{ij}(n)$ and $X_{ij}(n)$ are real and imaginary parts of impedance of branch n. VSI for all buses should be greater than 0 and lower than 1, of the network. Stability level of different buses can be measured and if some buses' VSI values are low, consequently MGs should be installed on them [31]. Our model objective function can be applied for each firefly of initial population using (1) to (35).

FA is an unconstrained solution method, therefore, violation of model constraints for each firefly can add a penalty to the objective function. FA fitness in iteration k can be calculated according to (36). In the proposed model, optimal solution is occurred when independent variables of planning and partitioning being in predefined operational limits, otherwise total

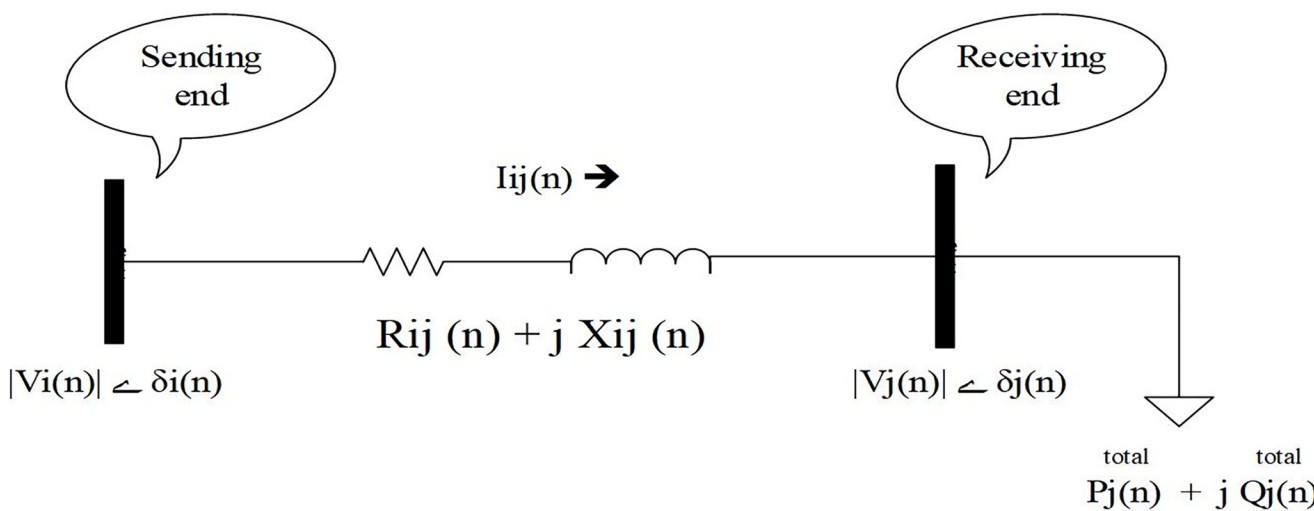

**Fig 3. Equivalent circuit of radial distribution feeder.**

penalty is imposed to fitness function as (37).

$$fitness(k, x_i^k) = \frac{1}{Of_{MGs}^{Planning}(k) + Penalties(k)} \tag{36}$$

$$Penalties(k) = \sum_{i=1}^{\substack{Operation \\ lim\ its}} (\left| IV_i - IV_i^{Max} \right| + \left| IV_i^{Min} - IV_i \right|) \times 10^6 \tag{37}$$

Index IV is an array of firefly position vector. For example $IV_i$ is equal to $IP_{CHP}$ (Bus 1) which indicated to optimal installation capacity on Bus 1, if more/less than maximum/minimum permissible values then a penalty will impose to it according to (37). The solution flowchart of the proposed model using FA is shown in Fig 4.

Firefly with maximum fitness is considered as best solution. The sum of all penalties due to violations from network and MMGs planning and operational constraints appears in the denominator of fitness. Editing the position vectors of fireflies is repeated during an iteration of the algorithm until gets to a predefined range. Fitness function determines each firefly's attractiveness and also, a firefly with better fitness absorbs all the fireflies which are in the neighborhood. Firefly luciferin value for iteration k is determined considering firefly fitness

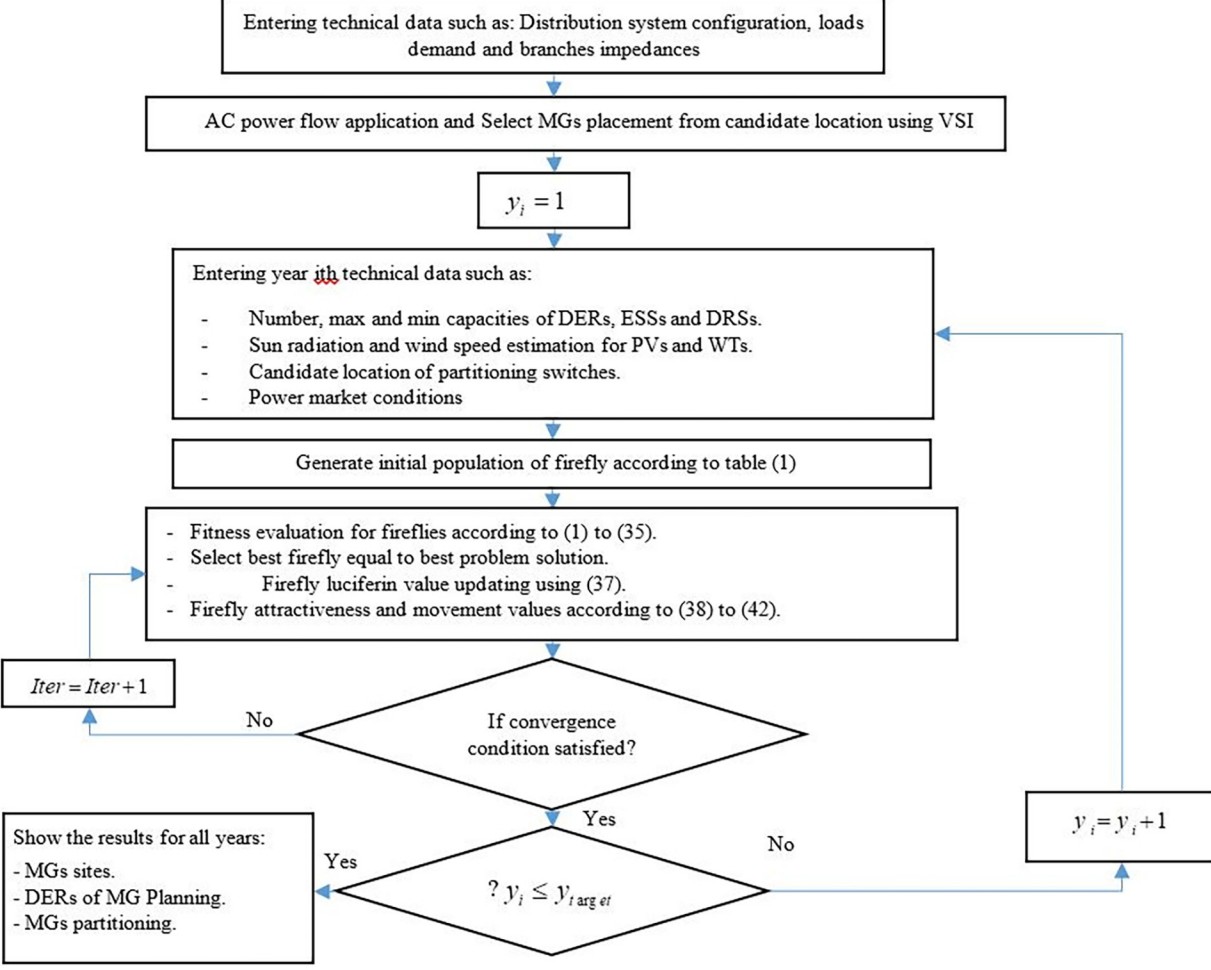

**Fig 4. The solution algorithm for planning and partitioning of multiple MGs using FA.**

iteration k and luciferin at the previous iteration according to relation (38).

$$L_i^k(t) = (1 - \rho).L_i^k(t - 1) + \gamma.fitness(x_i^k(t)) \tag{38}$$

Indexes $\rho$ and $\gamma$ are fixed numbers for modelling the gradual drop and fitness effect on firefly luciferin. During the movement phase, each firefly moves to one of its neighbors that has high luminosity with more luciferin, probabilistically. For $i^{th}$ firefly, movement probability toward neighbor firefly $j^{th}$ with high luminosity in iteration k is defined according to relation (39). Index $N_i$ is the collection of $i^{th}$ firefly neighbors. Time-discrete movement for a firefly can be written as (38) to (42).

$$P_{ij}^k(t) = \frac{L_j^k(t) - L_i^k(t)}{\sum\limits_{l \in N_i}(L_l^k(t) - L_i^k(t))} \tag{39}$$

$$x_i^k(t + 1) = (1 - \beta).x_i^k(t) + \beta.x_j^k(t) + \alpha.\varepsilon_i(t) \tag{40}$$

$$\beta^k = (1 - \beta_{min}).e^{-\gamma r_{ij}^k}(t) + \beta_{min} \tag{41}$$

$$r_{ij}^k = \frac{x_j^k(t) - x_i^k(t)}{|x_j^k(t) - x_i^k(t)|} \tag{42}$$

Indexes $\beta k$ and $r_{ij}{}^k$ are firefly attractiveness and movement values and also |.| shows soft Euclidian operator. Therefore, FA in the process of solving the MMGs planning problem works on firefly location instead of the problem parameters or variables.

## 4. Numerical studies

For the efficiency valuation of proposed model, numerical studies and simulations are performed on IEEE standard test distribution network and various case studies have been done. The apparent power for the distribution transformer in the substation is 20 MVA with a turn ratio of 63 kV/20 kV. The permissible range for change of voltage amplitude is between 0.95 and 1.05p.u, and also branch current is limited to 500 A. Costumers are modelled as static loads with fixed active and reactive power demands similar to reference [32].

Branch impedances are the same as this reference, too. Base values of apparent power and voltage magnitude for per unit calculation in the distribution system have been selected as 1000kVA and 20kv, respectively. The test distribution system has one main feeder and three sub-feeders. Main feeder includes Bus1 to Bus 18, sub-feeder1 includes Bus 19 to Buss 22, and subfeeder2 includes Bus 23 to Buss 25, and also sub-feeder3 includes Bus 26 to Buss 33. Candida location for TSs installation and MMGs investment have been depicted in the standard test distribution network as Fig 5. As can be seen in this figure, there are seven TSs which can be used for partitioning goal. Switch1 and Switch5 are installed belong to the main feeder between Bus 8-Bus 9, Bus 2-Bus 19 Bus 4-Bus 23, Bus6-Bus 26 and Bus 18-Bus 33, respectively.

Candida location for MG1 is defined between Bus19 and Bus22 through Switch2 and Switch7. Candida location for MG2 is defined between Bus 23 and Bus 25 through Switch 3, Switch6 and Switch7. Candida location for MG3 is defined between Bus26 and Bus33 through Switch4, Switch5 and Switch6. Candida location for MG4 is defined between Bus9 and Bus18 through Switch1 and Switch5. It is noted that the distribution network should be partitioned to MMGs so that the radial structure remains unchanged. The loss cost per kilowatt-hour is

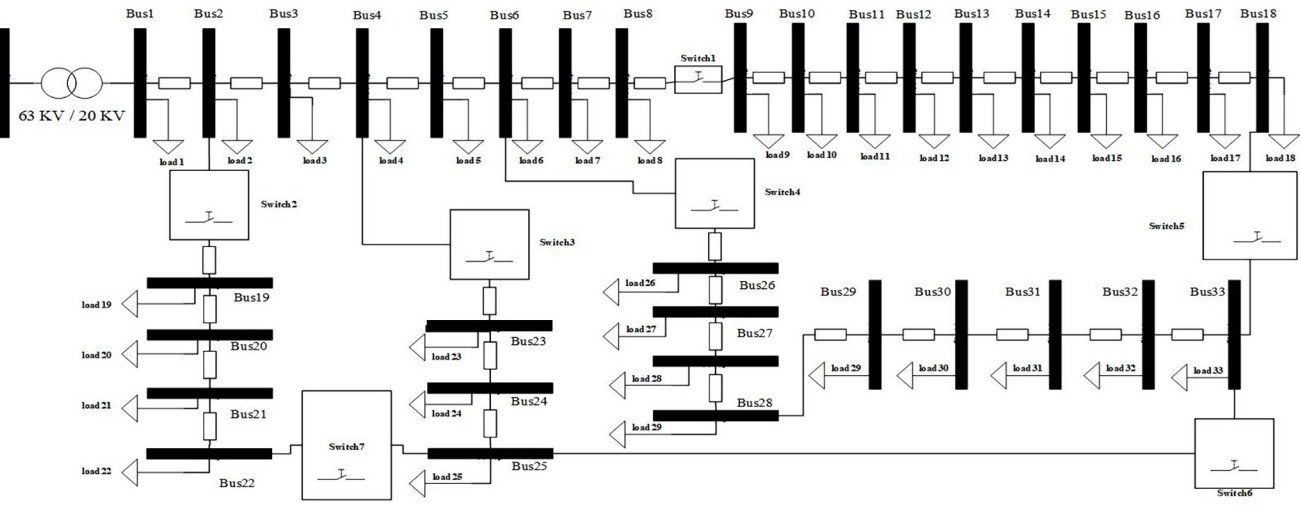

**Fig 5. IEEE standard test distribution network for MMGs planning and partitioning.**

equal to 0.4 dollars. The planning time horizon for under study distribution network is four years, and three loading factors have been considered for each year according to Fig 6.

Here, it is assumed that all MGs have one CHP, DRS and BESS, but, MG3 and MG4 will have installation possibility for one PV and one WT in addition to mentioned resources. Maximum allowable installation capacity of CHP in MG1, MG2, MG3 and MG4 are 25 kW, 20 kW, 15 kW and 15 kW, respectively. The coefficients of the cost function for them are according to Table 3. Allowable installation capacities of BESSs for all MGs are 15kw. Charging and discharging efficiencies of BESSs are equal to 0.95 and 0.99, respectively. WT3 and WT4 can be installed with maximum capacities of 10kw and 15kw, respectively. The cut-in, cut-out and nominal wind speed for WTs resources are assumed according to Table 4.

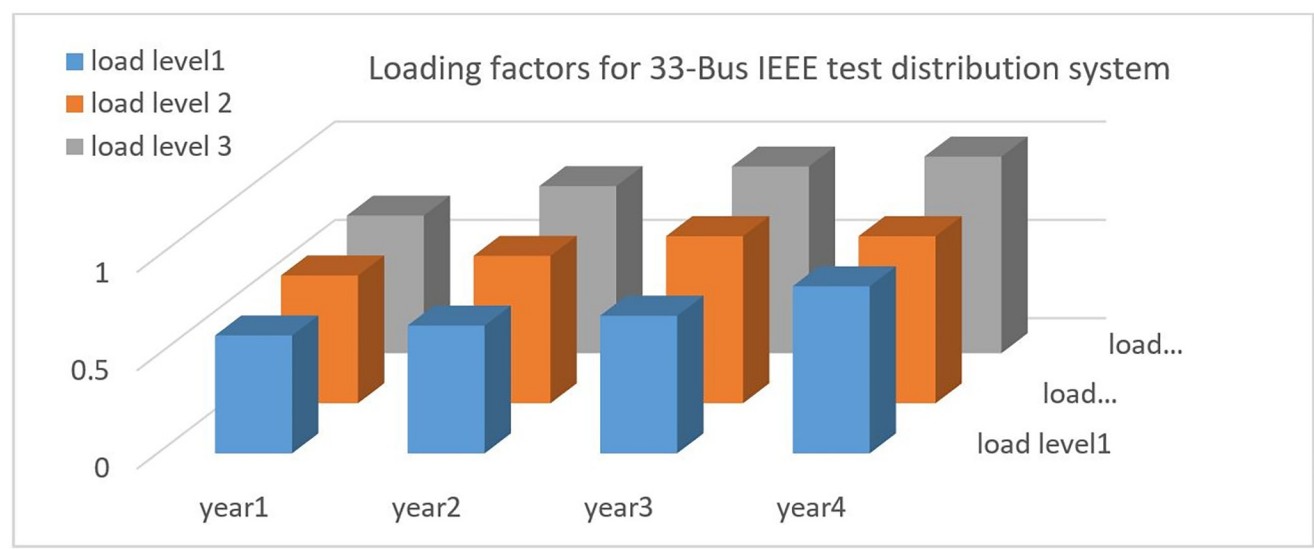

**Fig 6. Loading factors for planning process for each years.**

**Table 3. Cost function coefficients for all CHPs in MMGs.**

| Coefficients of Cost function | MG1 (CHP1) | MG2 (CHP2) | MG3 (CHP3) | MG4 (CHP4) |
|---|---|---|---|---|
| α | 3 | 2.5 | 2 | 1.5 |
| β | 0.63 | 0.44 | 0.25 | 0.16 |
| γ | 0.0003 | 0.00025 | 0.00035 | 0.00045 |

Normal prediction of wind speed at different load levels during each year of planning horizon have been shown according to Table 5.

Allowable capacities for PV3 and PV4 installation in MG3 and MG4 are considered equal to 15 kW. Scenarios of sun radiation and wind speed have been modelled by the probability-tree method according to Table 6. Various operation scenarios of PVs at different load levels during each year of the planning horizon have been shown in Table 7. The probability-tree method is used to model different scenarios of energy price in the electricity power market according to Table 8.

Three dimensional changes of energy prices at different load levels in planning time horizon are depicted in Fig 7.

Two case studies are arranged as below:

- MMGs planning without partitioning process for IEEE standard test distribution network.

- MMGs planning and partitioning process for IEEE standard test distribution network.

FA method is used to solve the mentioned optimization problem. Input parameters of FA algorithm such as firefly number, iteration, α and γ coefficients are set to 10, 200, 0.55, 0.85 and 1, respectively. ASUS laptop with 5 GHz core-i7 processor and 100 GB of external memory is used for the implementation of the proposed model. Model Codes are written in MATLAB software, and finally, numerical results have been expressed.

**Table 4. Technical specification of WTs for installation in MMGs.**

| Technical Specifications | $v_{WTi}^{cut-in}$ | $v_{WTi}^{cut-out}$ | $v_{WTi}^{rated}$ |
|---|---|---|---|
| MG3 (WT3) | 10 | 35 | 25 |
| MG4 (WT4) | 10 | 30 | 20 |

**Table 5. Normal wind speed at different load levels for planning process.**

| Wind Speed (m/s) | Planning Year | LL = 1 | LL = 2 | LL = 3 |
|---|---|---|---|---|
| MG3 (WT3) | $y_{li} = 1$ | 10.5 | 24 | 19 |
| | $y_{li} = 2$ | 11 | 20 | 18 |
| | $y_{li} = 3$ | 8 | 26 | 20 |
| | $y_{li} = 4$ | 10.25 | 13 | 15 |
| MG4 (WT4) | $y_{ji} = 1$ | 10.7 | 13 | 14 |
| | $y_{li} = 2$ | 10 | 12 | 15 |
| | $y_{li} = 3$ | 10 | 15 | 15 |
| | $y_{li} = 4$ | 11.5 | 14 | 13 |

**Table 6. Various scenarios of WTs operation at the planning horizon of MMGs.**

| Scenarios of Wind Speed | | | Scenario1 | Scenario2 | Scenario3 | Scenario4 | Scenario5 |
|---|---|---|---|---|---|---|---|
| **WT3** | year = 1 | Pr | 0.20 | 0.15 | 0.25 | 0.30 | 0.10 |
| | | AF | 0.93 | 0.92 | 0.83 | 0.92 | 0.94 |
| | year = 2 | Pr | 0.30 | 0.10 | 0.20 | 0.25 | 0.15 |
| | | AF | 0.98 | 0.95 | 0.88 | 0.91 | 0.95 |
| | year = 3 | Pr | 0.35 | 0.25 | 0.10 | 0.10 | 0.20 |
| | | AF | 0.93 | 0.97 | 0.88 | 0.95 | 0.85 |
| | year = 4 | Pr | 0.15 | 0.30 | 0.30 | 0.15 | 0.10 |
| | | AF | 0.94 | 0.95 | 0.85 | 0.95 | 0.88 |
| **WT4** | year = 1 | Pr | 0.23 | 0.12 | 0.25 | 0.25 | 0.15 |
| | | AF | 0.94 | 0.96 | 0.87 | 0.98 | 0.96 |
| | year = 2 | Pr | 0.30 | 0.10 | 0.20 | 0.20 | 0.20 |
| | | AF | 0.98 | 0.95 | 0.98 | 0.95 | 0.98 |
| | year = 3 | Pr | 0.30 | 0.20 | 0.15 | 0.15 | 0.20 |
| | | AF | 0.93 | 0.97 | 0.89 | 0.95 | 0.85 |
| | year = 4 | Pr | 0.10 | 0.30 | 0.25 | 0.20 | 0.15 |
| | | AF | 0.95 | 0.98 | 0.95 | 0.98 | 0.88 |

## 4.1 Base case study: Only MMGS planning process

The proposed model is run for the goal of MMGs Planning without Partitioning using FA method in MATLAB software. Convergence characteristics of fitness functions of FA for the four-year planning process are shown based on iteration according to Fig 8.

Here, it is assumed that all TSs have close status except switch5, switch6 and switch7. Therefore, simulation results show only the optimal sites and size of DERs for installation in MMGs according to Fig 9.

**Table 7. Various scenarios of PVs operation at the planning horizon of MMGs.**

| Scenarios of Sun Radiation | | | Scenario1 | Scenario2 | Scenario3 |
|---|---|---|---|---|---|
| **PV3** | year = 1 | Pr | 0.40 | 0.30 | 0.30 |
| | | AF | 0.98 | 0.92 | 0.84 |
| | year = 2 | Pr | 0.35 | 0.35 | 0.30 |
| | | AF | 0.95 | 0.94 | 0.85 |
| | year = 3 | Pr | 0.40 | 0.25 | 0.35 |
| | | AF | 0.90 | 0.93 | 0.85 |
| | year = 4 | Pr | 0.25 | 0.25 | 0.50 |
| | | AF | 0.92 | 0.95 | 0.84 |
| **PV4** | year = 1 | Pr | 0.45 | 0.25 | 0.30 |
| | | AF | 0.94 | 0.82 | 0.88 |
| | year = 2 | Pr | 0.25 | 0.50 | 0.25 |
| | | AF | 0.92 | 0.95 | 0.86 |
| | year = 3 | Pr | 0.30 | 0.35 | 0.35 |
| | | AF | 0.96 | 0.93 | 0.95 |
| | year = 4 | Pr | 0.35 | 0.15 | 0.50 |
| | | AF | 0.91 | 0.95 | 0.80 |

**Table 8. Scenarios of energy price in the electricity power market.**

| Price of Energy in the Power Market | | | Scenario1 | Scenario2 |
|---|---|---|---|---|
| Energy Price | year = 1 | Pr | 0.45 | 0.55 |
| | | AF | 0.90 | 0.80 |
| | year = 2 | Pr | 0.45 | 0.75 |
| | | AF | 0.88 | 0.85 |
| | year = 3 | Pr | 0.55 | 0.65 |
| | | AF | 0.96 | 0.94 |
| | year = 4 | Pr | 0.65 | 0.35 |
| | | AF | 0.95 | 0.90 |

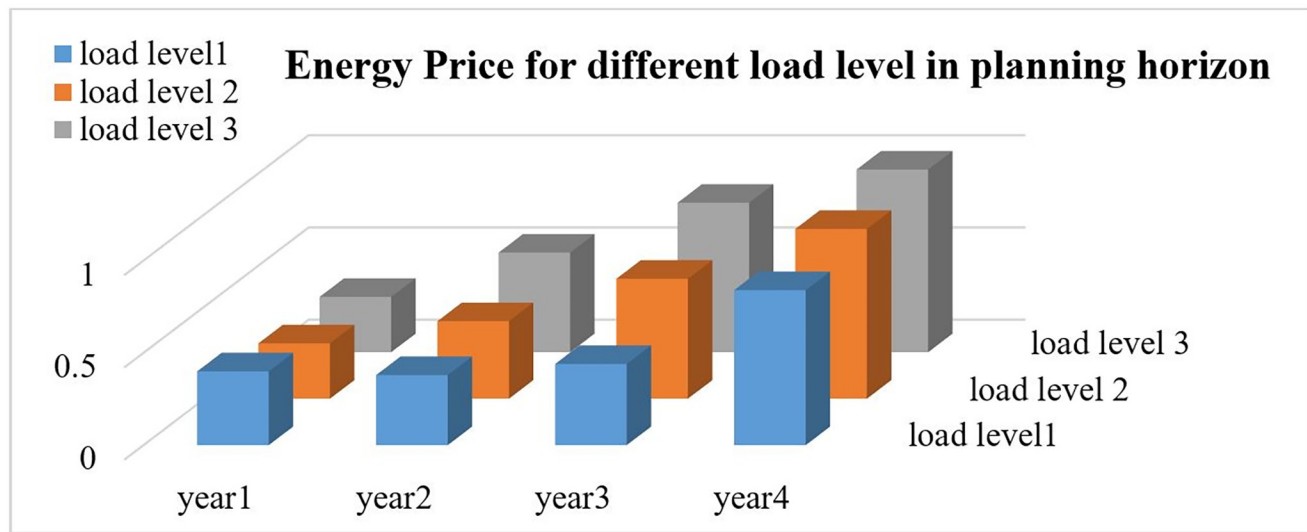

**Fig 7. Energy price at different load levels in planning horizon of MMGs.**

Simulation results for MG1 show that CHP1 with a capacity 10 kW at bus21, DRS1 with a capacity 12 kVar at Bus22 and BESS1 with capacity 1.75kw at Bus22 should be optimally installed. From simulation results, CHP2 with capacity 20 kW at bus24, DRS2 with capacity 10kVar at Bus25 and BESS2 with capacity 1.25 kW at Bus25 should be optimally installed in MG2. For MG3, CHP3 with capacity 10kw at bus28, DRS3 with capacity 12 kVar at Bus32 and BESS3 with capacity 2.5 kW at Bus33 should be optimally installed and from simulation results CHP4 with capacity 10kw at Bus13, DRS4 with capacity 12 kVar at Bus18 and BESS4 with capacity 3.25kw at Bus16 should be optimally installed in MG4. PV3 with a capacity 10 kW and PV4 with a capacity 15 kW are optimally installed on bus 33 and bus 16, respectively. WT3 with a capacity 5 kW and WT4 with a capacity 10 kW are installed on bus 30 and bus 14, respectively.

From the results of the base case study, it is understandable that a portion of active and reactive power load demands are supplied with DERs in MGs and remain energy requirement is supplied with a sub-transmission substation. In addition, optimal locations of PVs in MGs are beside BESSs to overcome their operation uncertainties. Predictably, voltage characteristics of the distribution system are improved after DERs investment and MGs placement. Fig 10

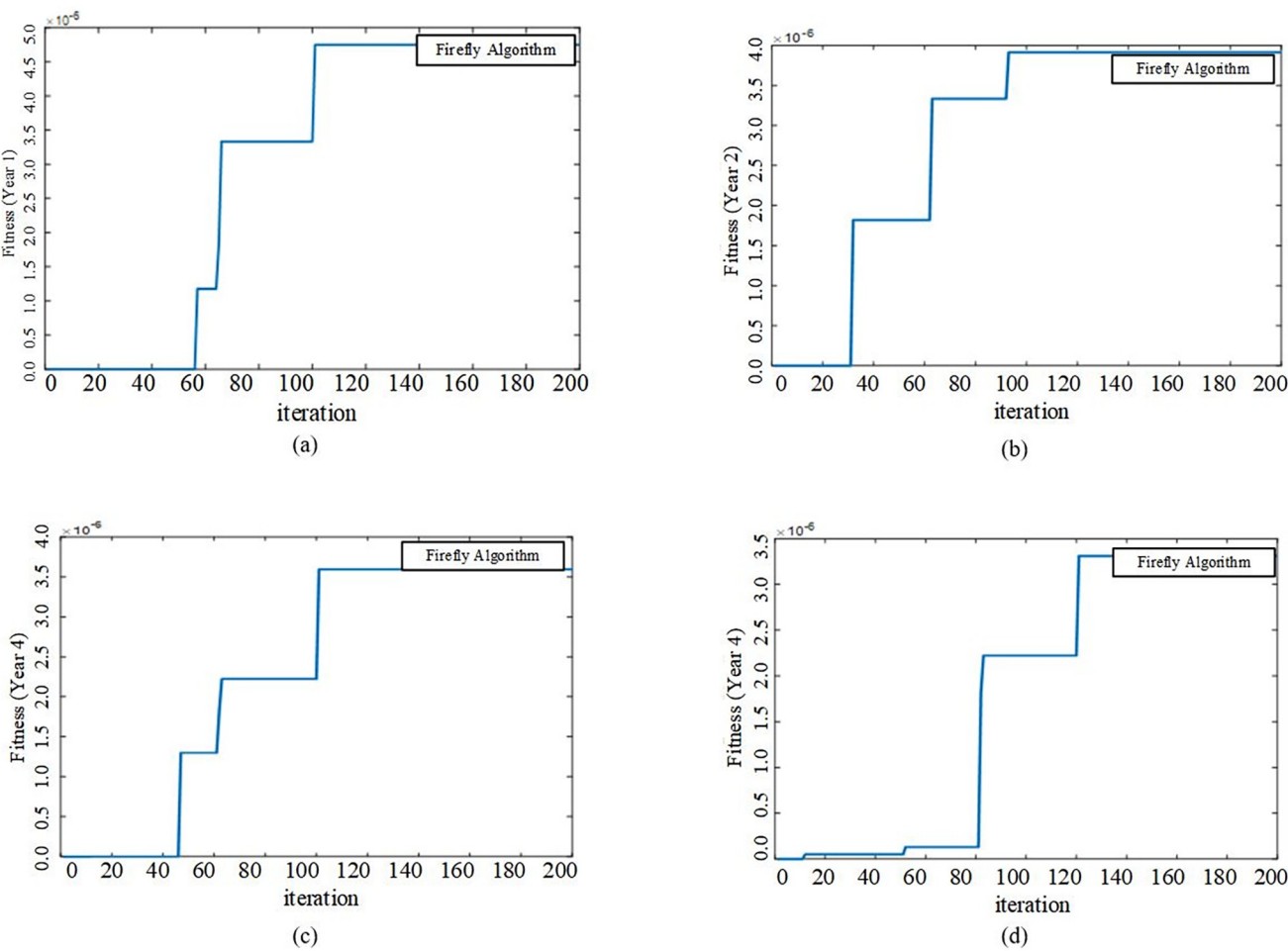

**Fig 8. Convergence characteristic of FA fitness functions for four year planning process.**

shows the voltage characteristic of the distribution system before and after MGs placement for various load levels.

It can concluded from the simulation results that the improvement of voltage profile on all feeders of the 33-Bus distribution system with optimal siting and sizing of DERs in all MGs in heavy loading situations is more than in light loading situations. In addition, simulation results show that bus 18 and bus 33 are detected as weak buses from the voltage stability point of view even after optimal MGs investment on the distribution network. With optimal planning of MMGs in a 33-Bus test distribution network, total energy loss will decrease by 2.25 MWh as compared to the absence of MMGs. At this condition, total MMGs risk is optimally evaluated as equal to 2415 dollars. Total planning and operation cost for year1, year2, year3 and year4 are evaluated 210555$, 255642 $, 287323 $ and 302101 $, respectively.

## 4.2 MMGS planning and partitioning

In this study case, proposed model is run for planning and partitioning of all MGs in distribution system using FA method in MATLAB software. All inputs remain unchanged and partitioning process is added to previous planning goal. Convergence characteristic of fitness functions for four years are shown as Fig 11.

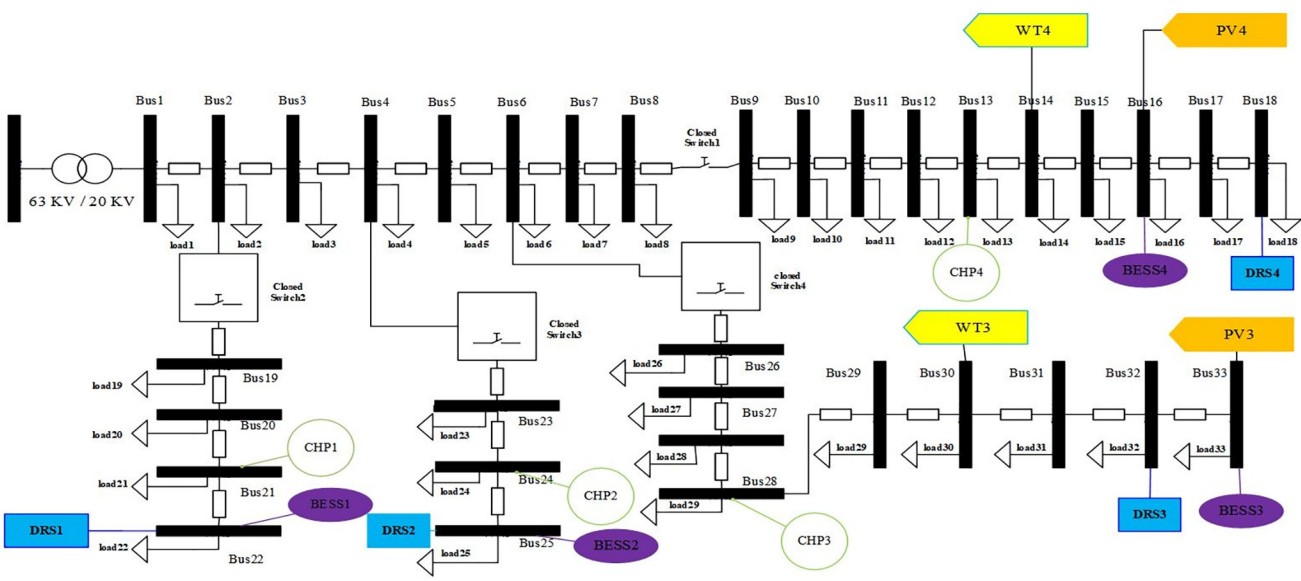

**Fig 9. Optimal placement of DERs for only MMGs planning.**

Similar to the previous case study, FA algorithm converges to optimal solution with good accuracy and speed. According to our expectation, the partitioning process with a change in distribution configuration can more effect on power flows among the feeders and decrease the total expected cost of system losses and reliability. The proposed model is resolved by the goal of the partitioning and planning of MMGs, simultaneously.

For new conditions, switch 1, switch 2, switch 4 and switch 7 have optimally close status and other switches have open status. The new configuration of MMGs causes to connection between MG2 and MG3. Therefore, it is expected that distribution system loss reduction, voltage stability and reliability indexes are improved after the sub-feeder shifts towards the beginning buses of the main feeder. Fig 12 shows the optimal results for the siting and sizing of DERs and also the siting of TSs.

Under new conditions, Bus 18, Bus 21, Bus 23 and Bus 29 are optimally selected as installation sites for CHP1, CHP2, CHP3 and CHP4 in MMGs, respectively. Optimal capacities of CHPs are determined 15kw, 15kw, 10kw and 15kw, respectively. DRSs are installed in MGs with the aim of reactive power compensation on connected buses and improving voltage magnitude in allowable ranges. In the new configuration of the distribution system DRS1, DRS2, DRS3 and DRS4 are optimally sited at bus 20, bus23, bus33 and bus15, respectively.

Optimal capacities of DRS1 to DRS4 are determined 10 kVar, 15 kVar, 15kVar and 10 kVar, respectively. PV3 and PV4 are located at bus 33 and bus17 with optimal capacities of 5kw and 10 kW, respectively. In addition, PV3 and PV4 are installed on bus 33 and bus17 with a capacity 5 kW with capacity 10 kW, respectively. Reconfiguration of the distribution system leads to a change in the optimal location of DERs in MMGs.

In the second case study, the sites of BESS1 with a capacity 4.5 kW and BESS2 with a capacity 5.5 kW remain unchanged as compared to the first case study, but, the sites of BESS3 with a capacity 6.55 kw and BESS4 with a capacity of 8.5 kw are optimally shifted from bus33 and bus15 to bus31 and bus 17, respectively. Numerical results reveal that BESSs have a more highlighted role in a new configuration of MMGs with more installation capacities besides WTs renewable resources. BESS3 is selected for facing power generation uncertainty related to

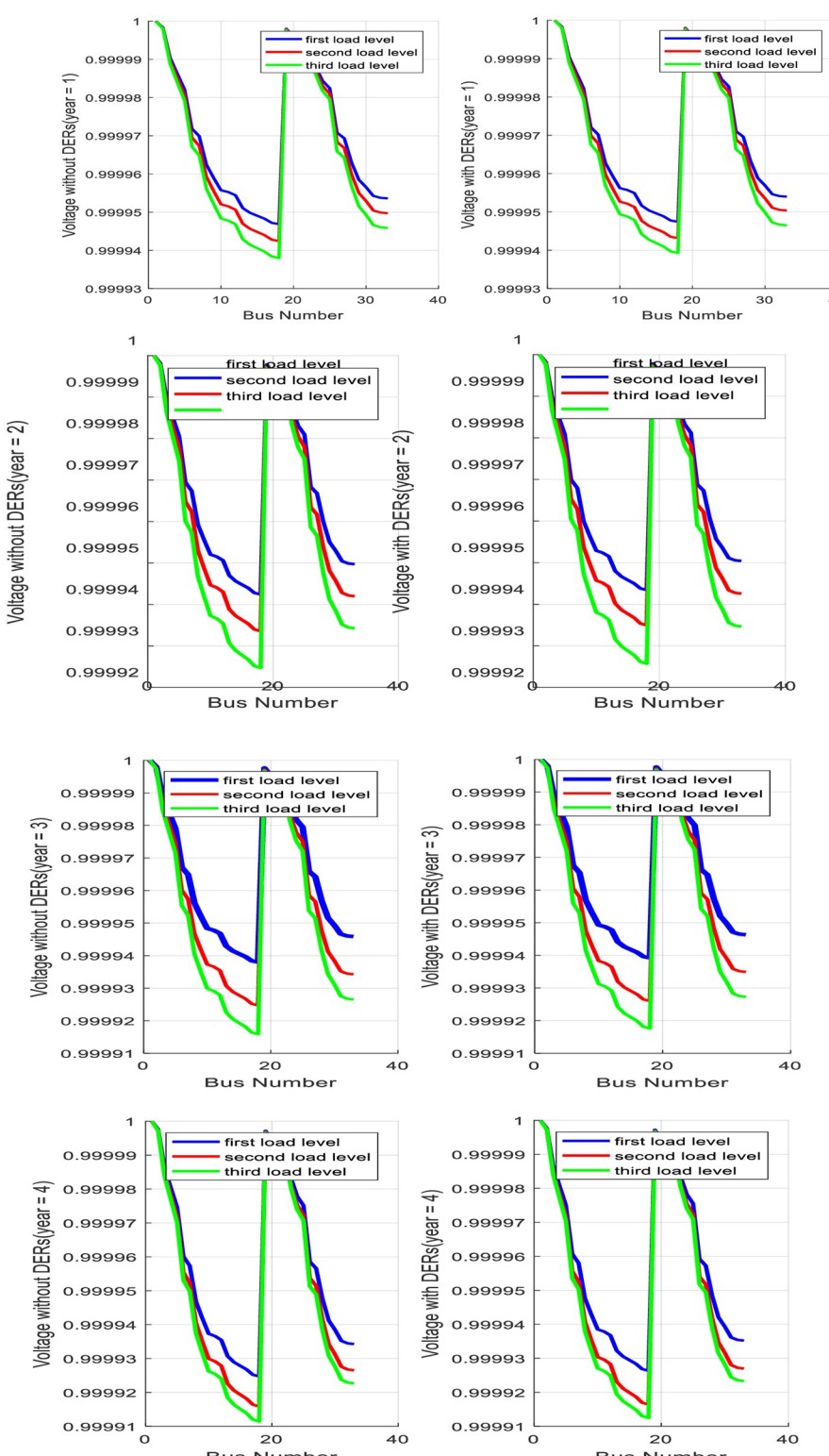

**Fig 10. Comparison of voltage profile with and without optimal planning of MMGs.**

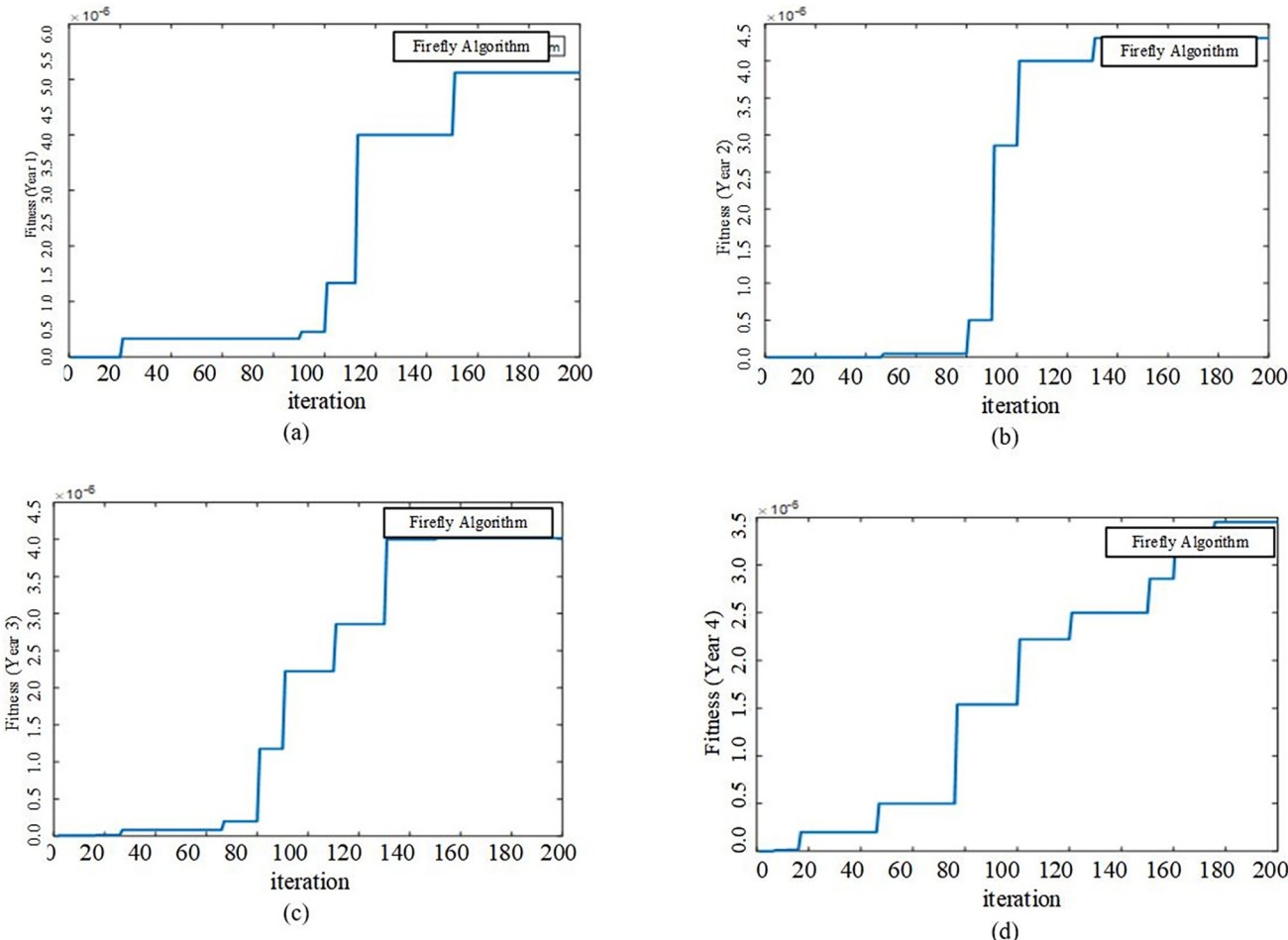

**Fig 11. Convergence characteristic of FA fitness functions for four years planning and partitioning process.**

WT3 with capacity 6.5kw in MG3 and BESS4 is selected for covering output uncertainty of WT4 with capacity 7kw in MG4. As the first case study, it is predictable that DERs placement on MMGs causes to improvement in voltage characteristics of under study distribution network. Fig 13 shows the simulation results for voltage characteristic of distribution network before and after the planning and partitioning of MMGs.

Under the new condition, distribution bus voltage values do not significantly change after planning and partitioning of MGs and Bus18 and Bus 33 are the weakest buses from a voltage stability point of view same as the previous study. The total expected energy not served is optimally determined at 2011.9 dollars. Two study comparisons show that 377.1 dollar decrement in total expected reliability cost. Total planning and operation cost for year1, year2, year3 and year4 are evaluated 195263 $, 232111 $, 248999 $ and 289564 $, respectively. Fig 14 shows the total expected loss of under-study distribution systems under various load levels when planning and partitioning are simultaneously applied.

The decrease in system expected loss after planning and partitioning of MGs is equal 1.12 MWh comparing the first case study. Then, simulation results show that appropriate planning and partitioning of MGs in the distribution system cause to decrease in total expected loss and reliability index during operation periods.

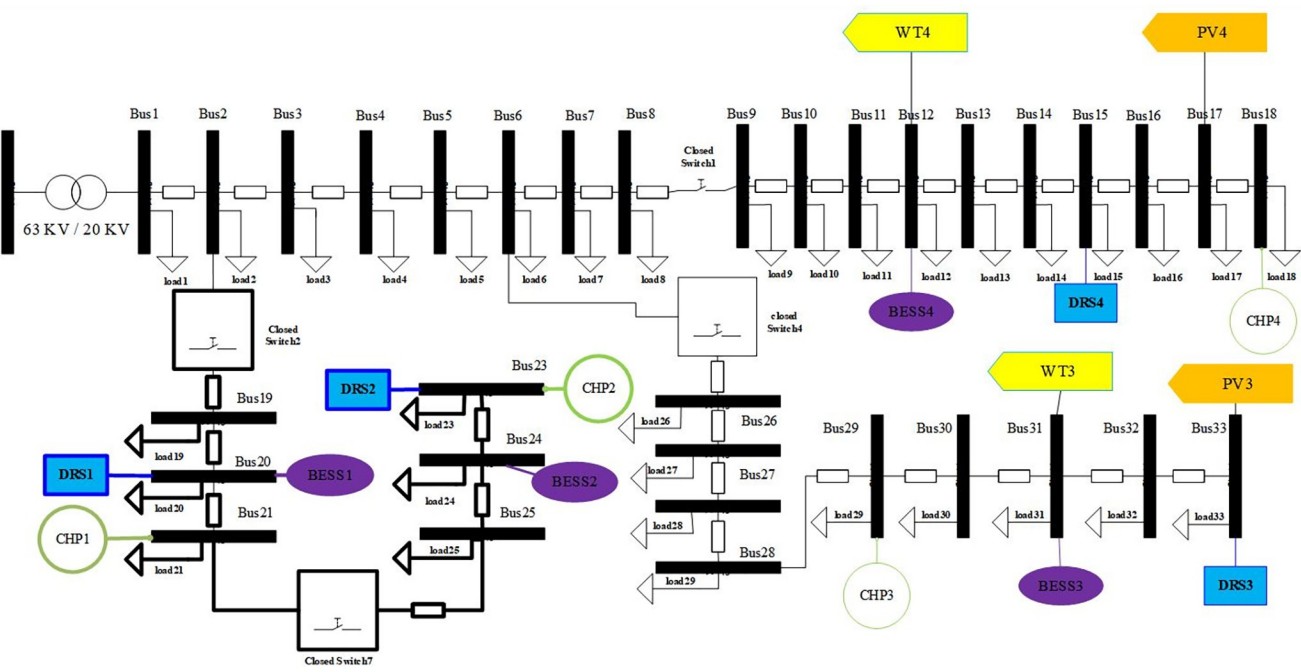

**Fig 12. Optimal siting and sizing of DERs in MMGs and TSs in distribution network (case study2).**

## 5. Results comparison

In this section, achieved simulation results have been compared for two case studies when the mentioned optimization problem is solved using FA, GA and PSO methods. Distribution network expected loss, total expected energy not served, total planning and operation cost for year1, year2, year3 and year4 have been compared in Table 9.

As can be seen in Table 9, the optimal number of iterations to achieve the convergence condition during the solution of case study1 using FA, GA, and PSO methods are between 90–125, 40–100 and 150–300, respectively. Under new conditions, iteration values during the solution of case study2 using three methods are between 130–180, 45–120 and 200–500, respectively. It can be concluded that GA and PSO methods have minimum and maximum iteration numbers to converge the optimal solution in both case studies, respectively. Generally, Simulation results show that GA solution results for two study cases are near to FA solution results.

## 6. Conclusion

In our article, a novel method is introduced with the goal of MMGs planning and partitioning about customer reliability issues. The main goal is minimizing the sum of equipment investment costs and expected cost of system loss and reliability and also expected cost of energy exchange. AC load flow is applied for studying the electrical situation of the distribution network. A proper voltage stability index has been applied to recognize the best investment sites of MGs in the distribution system. Operational risk is defined as expected energy not served considering uncertain operation scenarios of PVs and WTs renewable resources. For the effectiveness evaluation of the proposed model, numerical studies are applied to 33-bus distribution system including 4 micro-grids. In the absence of a partitioning option in the distribution system, appropriate siting and sizing of DERs in MGs lead to better voltage characteristics along

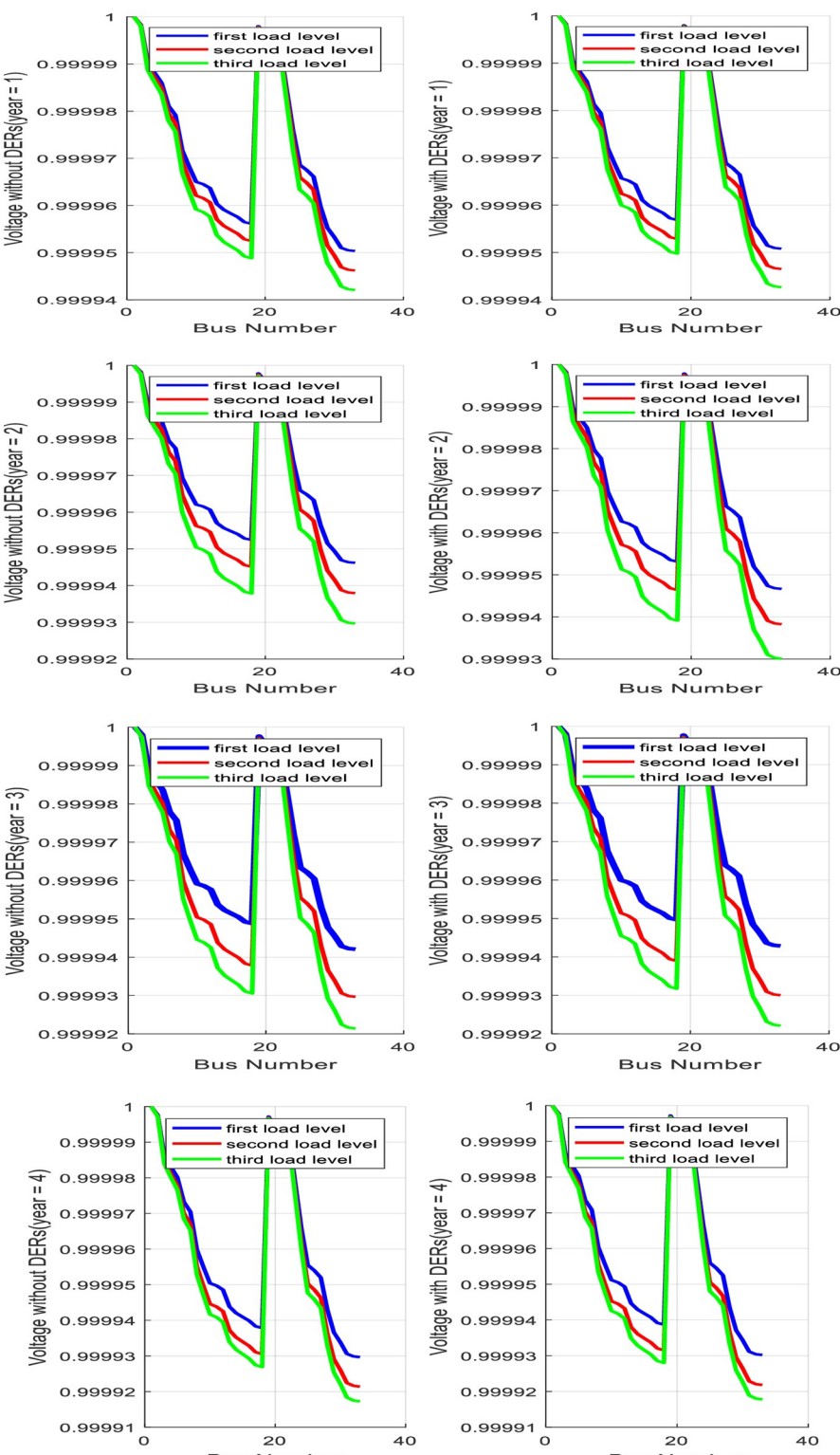

**Fig 13. Simulation results for voltage characteristic of distribution network before and after planning and partitioning of MMGs.**

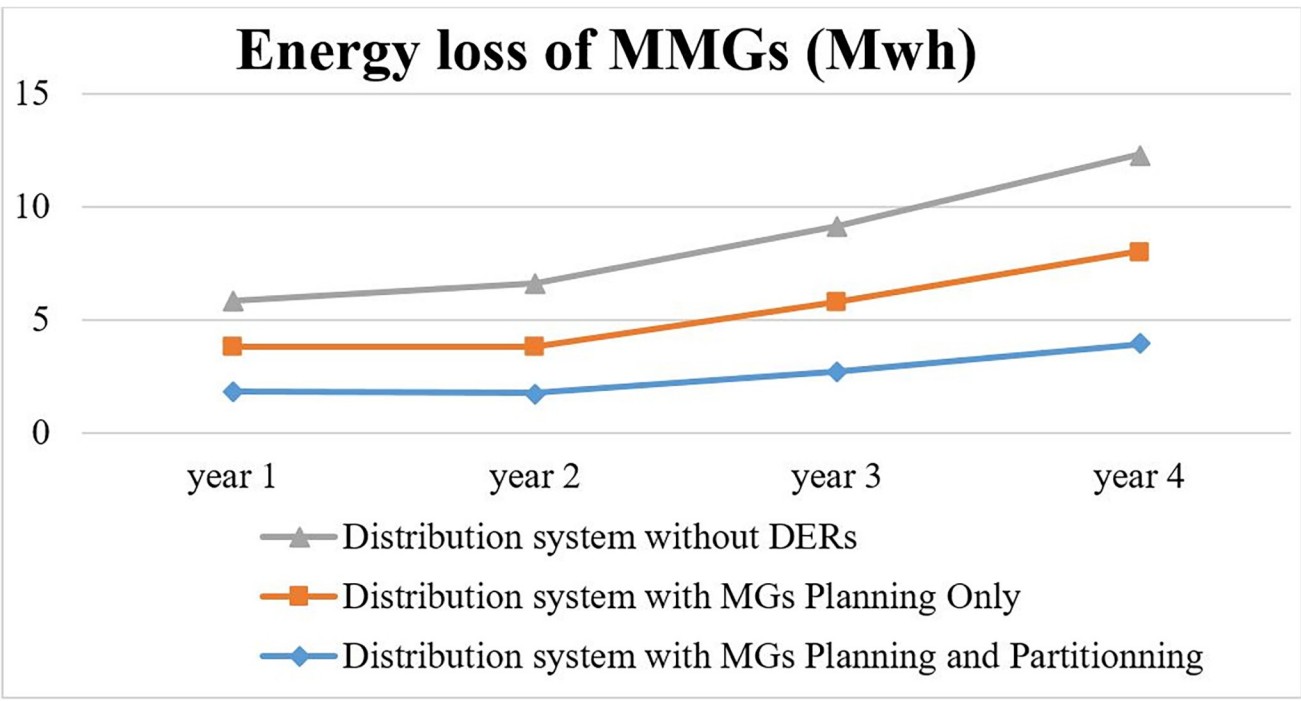

**Fig 14. Expected loss of distribution system under various load-level.**

**Table 9. Comparison results for two case studies with FA, GA and PSO.**

| CASE STUDIES | Solution Method | Iteration | Total Cost for Expected Loss (MWh) | Total cost for EENS ($) | Total planning and operation cost ($) | | | |
|---|---|---|---|---|---|---|---|---|
| | | | | | year1 | year2 | year3 | year4 |
| CASE 1 | FA | 90–125 | 11.215 | 2415.5 | 210555 | 255642 | 287323 | 302101 |
| | GA | 40–100 | 11.304 | 2550.8 | 210778 | 255845 | 287550 | 302280 |
| | PSO | 150–300 | 12.301 | 2760.5 | 212352 | 256005 | 288122 | 303556 |
| CASE 2 | FA | 130–180 | 10.170 | 2011.9 | 195263 | 232111 | 248999 | 289564 |
| | GA | 45–120 | 10.005 | 2089.5 | 195520 | 232650 | 249150 | 289785 |
| | PSO | 200–500 | 10.965 | 2155.8 | 196210 | 233195 | 249871 | 290002 |

the feeder specifically at low load and off-peak load. In the presence of a partitioning option besides planning of MGs, the optimal site and size of DERs in MGs are determined in such a way that all MGs connect to the buses which are located at the beginning of the feeder.

## Supporting information

**S1 File.**
(ZIP)

## Author Contributions

**Conceptualization:** Hamid Amini Khanavandi.

**Data curation:** Hamid Amini Khanavandi.

**Formal analysis:** Hamid Amini Khanavandi.

**Investigation:** Hamid Amini Khanavandi.

**Methodology:** Hamid Amini Khanavandi.

**Project administration:** Majid Gandomkar.

**Resources:** Hamid Amini Khanavandi.

**Software:** Hamid Amini Khanavandi.

**Supervision:** Majid Gandomkar, Javad Nikoukar.

**Validation:** Hamid Amini Khanavandi.

**Visualization:** Majid Gandomkar.

**Writing – original draft:** Hamid Amini Khanavandi.

**Writing – review & editing:** Majid Gandomkar, Javad Nikoukar.

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
