## [Decision Letter · Decision Letter 0]

3 May 2024

PONE-D-24-15840Optimal Planning and Partitioning of Multiple Distribution Micro-Grids Based on Reliability EvaluationPLOS ONE

Dear Dr. Gandomkar,

Thank you for submitting your manuscript to PLOS ONE. After careful consideration, we feel that it has merit but does not fully meet PLOS ONE’s publication criteria as it currently stands. Therefore, we invite you to submit a revised version of the manuscript that addresses the points raised during the review process.

We look forward to receiving your revised manuscript.

Kind regards,

Zhengmao Li

Academic Editor

PLOS ONE

Journal Requirements:

2. Please note that PLOS ONE has specific guidelines on code sharing for submissions in which author-generated code underpins the findings in the manuscript. In these cases, we expect all author-generated code to be made available without restrictions upon publication of the work. 

Please review our guidelines at https://journals.plos.org/plosone/s/materials-and-software-sharing#loc-sharing-code and ensure that your code is shared in a way that follows best practice and facilitates reproducibility and reuse.

3. We note that your Data Availability Statement is currently as follows: 

"All relevant data are within the manuscript and its Supporting Information files."

Reviewers' comments:

Reviewer's Responses to Questions

**Comments to the Author**

1. Is the manuscript technically sound, and do the data support the conclusions?

Reviewer #1: Yes

Reviewer #2: Yes

2. Has the statistical analysis been performed appropriately and rigorously? 

Reviewer #1: N/A

Reviewer #2: No

3. Have the authors made all data underlying the findings in their manuscript fully available?

Reviewer #1: No

Reviewer #2: Yes

4. Is the manuscript presented in an intelligible fashion and written in standard English?

Reviewer #1: No

Reviewer #2: No

5. Review Comments to the Author

Reviewer #1: This paper investigates the planning and partitioning of multiple distribution micro-grids. The topic is interesting. However, several areas that need attention:

1. The abstract needs to be restructured for clarity and to emphasize the novelty of the research. The current version of the abstract is too lengthy and may benefit from a more concise presentation to effectively communicate the key findings and contributions of the study.

2. The authors have reviewed previous studies that investigate multi-objective modeling. Could they clarify why multiple objectives were not considered in the present study?

3. The authors utilized stochastic optimization to handle uncertainty. It would be beneficial for them to compare stochastic optimization with existing uncertainty handling methods to highlight the advantages of their approach.

4. Adding a system structure in section 2 is recommended.

5. There is a notable need to enhance the quality of figures and tables presented in the paper, as they currently appear suboptimal.

6. The size of the equations is too large and should be adjusted for better readability.

7. The study lacks enough recent references, which is crucial for reflecting the latest advancements in the research topic. More references from the last two years are recommended.

Reviewer #2: This paper proposes a stochastic optimization for microgrid planning. I have the following comments for your consideration.

1. The abstract should be double-checked for typos, as the presence of "candida" seems glaringly incorrect.

2. The approach to handling uncertainty through the "probability tree" method appears to lack sufficient innovation.

3. It would be beneficial to further justify on the choice of genetic algorithms for problem-solving. Comparative analysis with established tools like Gurobi and CPLEX could strengthen the justification for the proposed calculation method.

4. The introduction would benefit from more recent references to enhance the review's currency.

5. Improvement in the font size and resolution of figures 2 and 6 is recommended for better readability.

6. Have you considered the operational stage to validate the investment decisions?

7. Could you demonstrate on how and at what stage the effect of operational risks is justified?

6. PLOS authors have the option to publish the peer review history of their article (what does this mean?). If published, this will include your full peer review and any attached files.

Reviewer #1: No

Reviewer #2: No

---

## [Author Response · Author response to Decision Letter 0]

23 May 2024

Thank you so much for sending the reviewers comments related to manuscript entitled “Optimal Planning and Partitioning of Multiple Distribution Micro-Grids Based on Reliability Evaluation”. We applied the reviewers’ comments to improve the quality of the paper. Responses to the reviewers’ comments are presented in the same order as the comments have been addressed.

---

## [Decision Letter · Decision Letter 1]

5 Jun 2024

Optimal Planning and Partitioning of Multiple Distribution Micro-Grids Based on Reliability Evaluation

PONE-D-24-15840R1

Dear Dr. Gandomkar,

We’re pleased to inform you that your manuscript has been judged scientifically suitable for publication and will be formally accepted for publication once it meets all outstanding technical requirements.

Kind regards,

Zhengmao Li

Academic Editor

PLOS ONE

Additional Editor Comments (optional):

Reviewers' comments:

Reviewer's Responses to Questions

**Comments to the Author**

1. If the authors have adequately addressed your comments raised in a previous round of review and you feel that this manuscript is now acceptable for publication, you may indicate that here to bypass the “Comments to the Author” section, enter your conflict of interest statement in the “Confidential to Editor” section, and submit your "Accept" recommendation.

Reviewer #1: All comments have been addressed

Reviewer #2: All comments have been addressed

2. Is the manuscript technically sound, and do the data support the conclusions?

Reviewer #1: Yes

Reviewer #2: Partly

3. Has the statistical analysis been performed appropriately and rigorously? 

Reviewer #1: Yes

Reviewer #2: Yes

4. Have the authors made all data underlying the findings in their manuscript fully available?

Reviewer #1: Yes

Reviewer #2: Yes

5. Is the manuscript presented in an intelligible fashion and written in standard English?

Reviewer #1: Yes

Reviewer #2: Yes

6. Review Comments to the Author

Reviewer #1: This paper investigates the planning and partitioning of multiple distribution micro-grids. The topic is interesting. My concerns are all responded. I have no further questions.

Reviewer #2: All of the comments and suggestions provided have been addressed, demonstrating the effort to enhance the quality of the paper. This paper is recommend for acceptance.

7. PLOS authors have the option to publish the peer review history of their article (what does this mean?). If published, this will include your full peer review and any attached files.

Reviewer #1: No

Reviewer #2: No

---

## [Editor Report · Acceptance letter]

20 Jun 2024

PONE-D-24-15840R1 

PLOS ONE

Dear Dr. Gandomkar, 

I'm pleased to inform you that your manuscript has been deemed suitable for publication in PLOS ONE. Congratulations! Your manuscript is now being handed over to our production team.

Kind regards, 

on behalf of

Dr Zhengmao Li 

Academic Editor

PLOS ONE